# Learning Generalizable Skill Policy with Data-Efficient Unsupervised RL

**Jongchan Park** [1]   **Seungjun Oh** [1]   **Seungho Baek** [1]   **Yusung Kim** [2][1]

## Abstract

Unsupervised Reinforcement Learning (URL) aims to pre-train scalable, skill-conditioned policies without extrinsic rewards, serving as a foundation for downstream control tasks. Despite recent progress, we argue that current off-policy URL methods are limited by two critical, overlooked bottlenecks: (1) non-stationary skill semantics and (2) brittle generalization. To address these challenges, we propose GenDa (Generalizable Data-efficient Agent), a unified framework for robust unsupervised reinforcement learning. First, we introduce a skill relabeling mechanism to mitigate non-stationarity and significantly improve data efficiency for pre-training. Second, we propose a Complementary Information Bottleneck (CIB), encouraging the learned skill policy to focus on ego-centric features and become robust to distribution shifts for downstream tasks. Through various experiments, we demonstrate that GenDa significantly enhances the scalability of URL with superior generalizability and data efficiency. Our code and videos are available at https://ihatebroccoli.github.io/official-GenDa/.

## 1. Introduction

Recent advances in unsupervised reinforcement learning (URL) have enabled the discovery of semantically distinct behaviors ("skills") from state transitions, without access to external reward signals (Gregor et al., 2016; Kim et al., 2021; Kamienny et al., 2022; Park et al., 2023; Yang et al., 2023). This paradigm aims to create a general-purpose foundation for control rather than just learning individual skills. It provides a pre-trained policy that readily adapts to diverse downstream tasks with minimal fine-tuning (Laskin et al.,

2021; Rajeswar et al., 2023). Achieving scalability in URL requires both efficiency during pre-training and robustness during transfer, ensuring the learned policy generalizes to varied downstream tasks.

Despite recent progress in URL, we argue that current state-of-the-art methods face two critical bottlenecks that hinder this scalability: (1) sample inefficiency arising from the overlooked non-stationary skill semantics, and (2) brittle generalization caused by overfitting to global context.

First, real-world interactions are costly, making data efficiency important. To maximize data efficiency, modern URL methods rely on off-policy algorithms that reuse past experiences stored in a replay buffer. However, this introduces a fatal flaw: **semantic drift**. In off-policy settings, each trajectory is collected under a randomly sampled skill $z$ and $z$-conditioned skill policy (Eysenbach et al., 2019; Sharma et al., 2020). The resulting $z$–trajectory pair is stored in a replay buffer and reused throughout training (Haarnoja et al., 2018; Laskin et al., 2022; Park et al., 2024; Zheng et al., 2025). As learning progresses, the behavioral trajectories induced by the same skill $z$ can change, because skill policy evolves across off-policy learning. The time-varying semantics of $z$ induce destabilization of the skill policy by providing stale samples.

Second, for a skill policy to be a scalable foundation for various downstream tasks, it must be robust to distribution shifts. A skill such as "walking forward" should be executable regardless of the global contextual information, such as environmental factors. However, in many prior works (Eysenbach et al., 2019; Park et al., 2024; Zheng et al., 2025), the observations used for training skill policies include global contextual information that is not essential for executing skills. When skill policies are conditioned on these global signals, the same skill may exhibit drastically different behaviors under slight shifts in coordinate distributions or become overfitted to specific global contexts. This brittleness limits the reusability of skills in downstream tasks and leads to fundamental generalization failures.

To address these challenges:

- We formalize two overlooked failure modes and provide theoretical and empirical evidence of their impact.

[1]Department of Artificial Intelligence, Sungkyunkwan University, Suwon, Republic of Korea [2]Department of Computer Science and Engineering, Sungkyunkwan University, Suwon, Republic of Korea. Correspondence to: Yusung Kim <yskim525@skku.edu>.

*Proceedings of the 43rd International Conference on Machine Learning*, Seoul, South Korea. PMLR 306, 2026. Copyright 2026 by the author(s).

- We introduce skill relabeling to mitigate the semantic drift in the previous off-policy URL. We also propose a novel regularizer to ensure skill diversity under the mutual information objective, effectively differentiating our skill relabeling framework from prior literature in other fields (Andrychowicz et al., 2017).

- We propose a Complementary Information Bottleneck (CIB), a drop-in module compatible with an unsupervised framework. CIB learns an embedding that prevents the policy from exploiting global contextual information, improving skill execution consistency under distribution shifts.

We evaluate our approach on a diverse set of state and pixel benchmarks, demonstrating superior data efficiency and skill-policy generalization compared to prior methods. Notably, we also consider high-dimensional state environments, where existing approaches fail to discover meaningful skills, while our method successfully learns coherent and reusable skill policies.

## 2. Preliminaries

**Unsupervised skill discovery** We follow the settings of prior work (Park et al., 2024), a controlled Markov process without a reward function. MDP is defined as $\mathcal{M} = (\mathcal{S}, \mathcal{A}, \mu, p)$. $\mathcal{S}$ represents the state space, $\mathcal{A}$ represents the action space, $\mu \in \Delta(\mathcal{S})$ represents the initial state distribution, and $p: \mathcal{S} \times \mathcal{A} \to \Delta(\mathcal{S})$ represents the transition dynamics kernel. We also consider a discrete or continuous set of latent vectors $z \in \mathcal{Z}$ with a latent-conditioned skill policy $\pi(a \mid s, z)$.

In the terminology of unsupervised skill discovery, the latent skill vector $z$ is sampled from the prior distribution $p(z)$. Then skill policy $\pi(a \mid s, z)$ rolls out a trajectory $\tau = (s_0, a_0, s_1, a_1, ..., s_T)$ with a fixed $z$ for the entire episode. In this setting, the main goal is to learn diverse and useful behaviors $\pi(a \mid s, z)$ from scratch without guidance (e.g., data, prior knowledge, and supervision).

We follow the representation objective of the metric-aware unsupervised skill discovery (Park et al., 2024) with latent mapping function $\phi(s)$ to inject semantic meaning in the skill vector $z \in \mathcal{Z}$:

$$\sup_{\pi,\phi} \mathbb{E}_p(\tau, z) \left[ \sum_{t=0}^{T-1} (\phi(s_{t+1}) - \phi(s_t))^\top z \right] \quad (1)$$

$$\text{s.t.} \|\phi(s_{t+1}) - \phi(s_t)\|_2 \leq 1, \forall(s_t, s_{t+1}) \in \mathcal{S}_{adj}$$

**Downstream application with pre-trained skill policy** A pre-trained skill policy $\pi$ can be adapted to downstream tasks. A hierarchical controller $\pi^h(z|s)$ samples from the set of learned skills and delivers $z$ to the (frozen) skill policy to maximize the downstream reward.

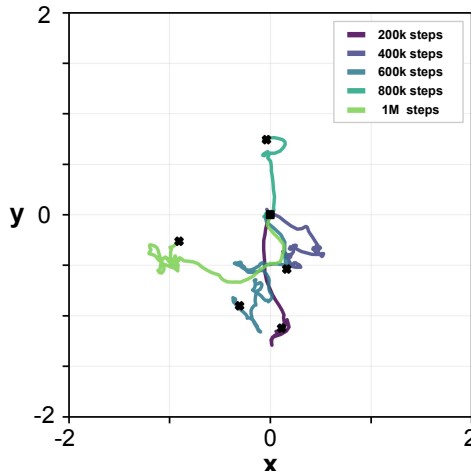

*Figure 1.* **Semantic drift in prior algorithm (Park et al., 2024).** This figure shows trajectories collected by the skill policy using the same skill vector z, illustrating that different behavioral trajectories can be observed for the same z during training. In standard off-policy URL, these trajectories are all stored in the replay buffer and repeatedly reused for representation learning. As a result, a one-to-many mapping between z and trajectories emerges, which destabilizes representation learning and leads to semantic drift.

## 3. Problem Statements

We formally identify the two core challenges preventing off-policy URL from achieving scalability.

### 3.1. Sample Inefficiency from Semantically Ungrounded Skill Rollouts

In the off-policy URL setting, the agent collects a trajectory $\tau$ conditioned on a sampled skill $z \sim p(z)$. The transition tuple, including the skill label $(\tau, z_{roll})$, is stored in a replay buffer $\mathcal{B}$. The critical issue lies in the dual role of $z$: it serves as both the input skill for the policy $\pi(a|s, z)$ and the target label for the representation $\phi(s)$.

Unlike standard RL, where the reward function is static, in URL, the semantic meaning of $z$ is defined by the representation $\phi$, which is continuously evolving. Let $\phi_t$ denote the representation at training step $t$. A trajectory $\tau$ generated at time $t_{old}$ with skill $z$ satisfied the relationship $z \approx \phi_{t_{old}}(\tau)$. However, at the current step $t_{curr}$, the updated representation $\phi_{t_{curr}}$ may map this trajectory to a completely different point in the latent space.

Most existing methods ignore this drift, updating the current policy using the stale pair $(\tau, z)$ from the buffer. This creates a semantic misalignment: the policy is penalized for failing to generate $z$ according to the current $\phi_{t_{curr}}$, even though the trajectory was correct under $\phi_{t_{old}}$. Consequently, no meaningful correction occurs until the agent gathers enough "fresh" samples to effectively displace or outweigh the stale

data. This acts as a source of high-variance noise in the gradient estimation. Consequently, increasing the sample reuse rate (i.e., high UTD ratio)—which typically improves efficiency in off-policy RL—ironically worsens instability in URL by overfitting to this persistent label noise.

Figure 1 illustrates that the same skill z can be associated with different trajectories depending on the training stage of $\phi$ and $\pi$, highlighting a potentially critical failure mode in the off-policy URL framework.

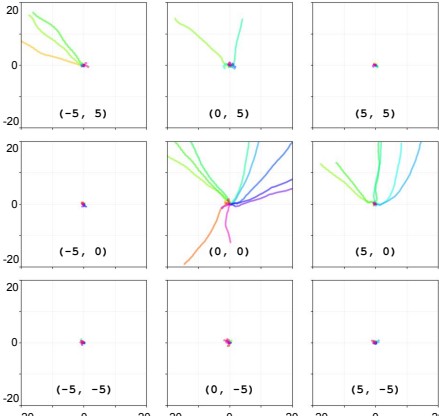

*Figure 2.* **Overfitting to xy-coordinates (global context).** An offset (a,b) indicates that the agent's initial xy position is set to (a,b) at evaluation time. During training, the agent always starts from (0,0), so this setup evaluates whether a given skill z produces consistent behavior under shifted initial conditions. Curves of the same color represent trajectories generated by the same skill z. When different offsets are applied, the skill policy (Park et al., 2024) fails to reproduce consistent trajectories, providing clear evidence of brittle generalization.

### 3.2. Lack of Skill-Policy Generalization under Downstream Tasks

A second limitation concerns the limited generalization capability of the learned skill policy. In URL, skill policies are typically intended to function as consistent local behavioral primitives that respond to local state changes, directed by the given skill's guidance. This property is a key prerequisite for reusing skill policy across diverse initial conditions and downstream environments.

In most prior works, however, skill policies are often conditioned directly on the full state observation. Such observations include global contextual information (e.g., absolute positions or visual background cues) that is not essential for executing local behaviors. When the policy is conditioned on this information, skill execution becomes entangled with the global context, weakening the intended locality of the learned behaviors.

This entanglement makes the policy fragile under distri-

bution shifts. Changes in the global context between pre-training and deployment cause the same latent z to produce unreliable behaviors, significantly hindering downstream performance.

In Figure 2, we show a failure mode of the prior algorithm where conditioning on full state observation fails to execute reliably on varying global contexts. This failure highlights that maximizing state coverage during pre-training is a necessary but insufficient condition for scalability; the learned skills must also be disentangled from the specific global context in which they were discovered.

## 4. Method

We propose GenDa (Generalizable Data-efficient Agent), a unified framework designed to train a scalable skill foundation for control. Unlike prior works that treat skill discovery as a static optimization problem, GenDa addresses the dynamic nature of off-policy learning and structural requirements for generalization. We introduce (a) skill relabeling to dynamically align past experiences with the evolving latent space, and (b) Complementary Information Bottleneck (CIB) to structurally encourage the policy to learn ego-centric, reusable behaviors. Our approach is based on the previous state-of-the-art algorithm (Park et al., 2024).

### 4.1. Relabeling for Skill Representation Learning

#### 4.1.1. RELABELING WITH THE CURRENT LATENT INTERPRETATION.

Standard off-policy URL methods suffer from **semantic drift**, where the skill labels z stored in the replay buffer become stale as the representation $\phi$ evolves. Updating the policy with these stale labels injects high-variance noise into the learning process, thereby limiting the data efficiency in the off-policy setting. To overcome this, we introduce skill relabeling, which treats past trajectories not as fixed $(\tau, z)$ pairs, but as flexible experiences that can be re-interpreted.

**Relabeling (z-step)** To correct the semantic inconsistency caused by fixed rollout-time $z_{roll}$ in the replay buffer, we need to define a new label z that is consistent with the current $\phi$. To define that z, we revisit the telescoping sum in Equation 1. For an episode $\tau = (s_0, a_0, s_1, ..., s_T)$, the telescoping sum can be written as $(\phi(s_T) - \phi(s_0))^\top z$, which means $\sum_{t=0}^{T-1} (\phi(s_{t+1}) - \phi(s_t))^\top z$. We can define this telescoping sum as delivering the most appropriate direction vector to explain $\tau$ with on-policy $\phi$.

Instead of using $z_{roll}$, we compute $z_{\text{relab}}$ from the current

latent function:

$$z_{\text{relab}}(\tau) = \text{unit}(\phi(s_T) - \phi(s_0)),$$
$$\text{unit}(x) := \frac{x}{\max(\|x\|_2, \epsilon)}. \tag{2}$$

where $\epsilon$ is a very small positive scalar.

**Optimization ($\phi$-step)** We then learn with the objective in equation 1 by replacing $z$ with $z_{\text{relab}}$ while stopping gradients:

$$\max_{\phi} \mathbb{E}_{\tau \sim \mathcal{B}} \left[ \sum_{t=0}^{T-1} (\phi(s_{t+1}) - \phi(s_t))^{\top} z_{\text{relab}} \right]$$
$$\text{s.t.} \|\phi(s_{t+1}) - \phi(s_t)\|_2 \leq 1, \forall (s_t, s_{t+1}) \in \mathcal{S}_{adj} \tag{3}$$

Iterating ($z$-step $\to$ $\phi$-step) mitigates the gradient conflicts caused by the fixed $z_{\text{roll}}$, thereby resolving the aforementioned semantic drift. We demonstrate that our iterative update is mathematically robust and structurally stable at each step (Details in Appendix A).

$$z_{relab} = \text{unit}(\phi_{\text{tgt}}(s_T) - \phi_{\text{tgt}}(s_0)). \tag{4}$$

**Implementation details of skill relabeling** In practice, to reduce training instability, we compute the relabeled targets using an exponential moving average (EMA) network (Schwarzer et al., 2021), as shown in Equation 4.

### 4.1.2. PREVENTING COLLAPSE VIA A UNIFORMITY REGULARIZER.

In prior works utilizing a fixed $z_{roll}$ label, the distribution of $z_{roll}$ is naturally guaranteed to follow a random distribution $p(z)$, which is fixed after sampling. Consequently, the entropy term $H(z)$ becomes a trivial constant in the objective for $\phi$.

While relabeling stabilizes learning, naively maximizing alignment with re-assigned labels can lead to **representation collapse**, where $\phi$ maps all states to a narrow region of the latent space to trivially satisfy the objective (Equation 3). Since the distribution of the relabeled skill $z_{relab}$ is not guaranteed to follow $p(z)$, to prevent such collapse, we introduce a contrastive-style *uniformity* regularizer to explicitly maximize $H(z_{relab})$.

Let $v_i = \text{unit}\left(\phi(s_T^{(i)}) - \phi(s_0^{(i)})\right)$ be the episodic direction for episode $i$ in a mini-batch. We define

$$\mathcal{L}_{\text{unif}} = \frac{1}{B} \sum_{i=1}^{B} \log \sum_{\substack{j=1 \\ j \neq i}}^{B} \exp\left(v_i^{\top} v_j\right), \tag{5}$$

where $B$ is the number of samples in a mini-batch. We *minimize* $\mathcal{L}_{\text{unif}}$, which improves the lower bound of

$H(z_{relab})$ (van den Oord et al., 2018). By encouraging the skill vectors to be uniformly distributed on the hypersphere, we ensure that the agent discovers a diverse repertoire of behaviors even while continuously redefining what those behaviors are.

### 4.1.3. FINAL REPRESENTATION OBJECTIVE.

Let $\mathcal{L}_{\text{base}}$ denote the representation loss for the objective in Equation 3. Our final representation learning loss is

$$\mathcal{L}_{\phi} = \mathcal{L}_{\text{base}} + \beta \, \mathcal{L}_{\text{unif}} \tag{6}$$

where $\beta$ is a scalar coefficient (we use $\beta = 1.0$ for all environments). Figure 9 (in Appendix) depicts that our uniformity term can handle the induced bias well. In Appendix B.2, we provide a theoretical analysis demonstrating that relabeling stabilizes the representation learning process in an off-policy setting.

### 4.2. Relabeling for Skill Policy Learning

In the policy learning phase, the intrinsic reward function

$$r(\phi, z) = (\phi(s') - \phi(s))^{\top} z \tag{7}$$

evaluates how well a transition $s \to s'$ follows a conditioned $z$ in the latent space based on the magnitude and direction of $\Delta\phi$. Prior methods use only fixed $z_{\text{roll}}$ as a condition of the intrinsic reward function, which limits the privilege of the off-policy setting.

We instead relabel each transition with multiple targets derived from diverse state pairs within the same trajectory to enrich the off-policy reward signal by exploiting the intrinsic reward in Equation 7.

For a transition at time $t$, we sample a horizon $c > 0$ and define:

$$z_c = \text{unit}(\phi(s_{t+c}) - \phi(s_t)). \tag{8}$$

We mix $z_c$ with the $z_{\text{relab}}$ and the original rollout label $z_{\text{roll}}$ (details in Appendix C) as commonly done in relabeling to preserve on-policy semantics in practice (Andrychowicz et al., 2017). This enables the policy/value function to learn from a richer set of $(s, s', z)$ configurations than those directly experienced, improving bootstrapping.

### 4.3. Complementary Information Bottleneck (CIB)

A pre-trained foundational skill policy $\pi(a \mid s, z)$ is desired to execute pre-trained skills for diverse applications, where distributional shifts in global contextual information frequently occur. However, previous algorithms use the full state as an observation that leads to unintended overfitting on the global context. To disentangle this connection, we propose the Complementary Information Bottleneck (CIB).

---

**Algorithm 1** GenDa Unsupervised Skill Discovery

---

1: Initialize skill policy $\pi(a \mid \ell, z)$, representation function $\phi$, EMA of representation function $\phi_{tgt}$, CIB $q_\psi(\ell \mid s)$, Lagrange multiplier $\lambda$, replay buffer $\mathcal{B}$
2: **for** $i \leftarrow 1$ to (# epochs) **do**
3:    **for** $j \leftarrow 1$ to (# episodes per epoch) **do**
4:       Sample skill $z \sim p(z)$
5:       Collect trajectory $\tau$ using $\pi(a \mid \ell, z)$ with $\ell \sim q_\psi(\cdot \mid s)$, and store to $\mathcal{B}$
6:    **end for**
7:    **for** $k \leftarrow 1$ to (# gradient steps per epoch) **do**
8:       Sample a batch $x = (s_t, a_t, s_{t+1}, s_{t+c}, s_0, s_T, z_{roll})$ from $\mathcal{B}$
9:       $x_{relab} \leftarrow$ Relabel $z_{roll}$ in $x$ to $z_{relab}$ using Eq. 4
10:      $x_c \leftarrow$ Relabel $z_{roll}$ in $x$ to $z_c$ using Eq. 8
11:      Update representation $\phi(s)$ and Lagrange multiplier $\lambda$ using $x_{relab}$ with Eq. 6
12:      Create mixed batch: $x_{mix} \leftarrow x \cup x_{relab} \cup x_c$
13:      Update skill policy $\pi(a \mid \ell, z)$ using SAC (Haarnoja et al., 2018) with $x_{mix}$
14:      Update CIB to maximize Eq. 9
15:      Update EMA target: $\phi_{tgt} \leftarrow \alpha\phi_{tgt} + (1 - \alpha)\phi$
16:    **end for**
17: **end for**

---

### 4.3.1. KEY IDEA.

Metric-aware skill discovery methods encourage the latent space $\phi$ to capture temporal distances. Often, the global contextual information is desired for the $\phi$ latent space, because this objective promotes the agent to capture the extensive "temporal" manifolds in the state space (Park et al., 2024). We desire to make the policy rely on the skill without entangling global contextual information in the raw state. To address this, we propose substituting the raw policy observations with a learned embedding that is *complementary* to $\phi(s)$.

We train an encoder $q_\psi(\ell \mid s)$ that produces an embedding $\ell$, together with a decoder $p_\omega(s \mid \ell, sg(\phi(s)))$, by maximizing a variational information bottleneck (Alemi et al., 2017) style objective:

$$\mathcal{J}_{\text{CIB}} = I((\ell, \phi(s)); s) - \eta I(\ell; s) \qquad (9)$$

where $\eta$ is a scalar coefficient (we use $\eta = 0.1$ for all environments). Intuitively, the decoder can use $\phi(s)$ to reconstruct whatever information $\phi$ already captures; thus, $q_\psi(\ell \mid s)$ is encouraged to preserve the remaining information needed to reconstruct $s$, making $\ell$ complementary to $\phi(s)$.

### 4.3.2. SKILL POLICY WITH CIB.

We replace the raw state input of the skill policy with the CIB embedding $\ell$:

$$\pi(a \mid s, z) \quad \longrightarrow \quad \pi(a \mid \ell, z), \qquad \ell \sim q_\psi(\ell \mid s). \quad (10)$$

The CIB is used consistently during data collection, training, and downstream deployment. The CIB encoder only changes the observation channel the policy conditions on, improving robustness to shifts in global contextual information.

## 5. Experiments

We design our experiments to verify whether GenDa successfully overcomes the scalability bottlenecks identified in Sec. 3. Specifically, we investigate the following questions:

1. **Data Efficiency** *Does skill relabeling enhance the data efficiency of the pre-training phase by mitigating the non-stationarity of off-policy learning?*

2. **Generalization** *Does disentangling global context via CIB produce robust skills that generalize to unseen initial conditions and downstream tasks?*

3. **Scalability** *Can GenDa serve as a better foundation policy for challenging, high-dimensional control problems compared to state-of-the-art baselines?*

As baselines, we include **METRA** (Park et al., 2024) and **CSF** (Zheng et al., 2025), which are state-of-the-art *metric-aware* URL methods, as well as **DIAYN** (Eysenbach et al., 2019) and **CIC** (Laskin et al., 2022), which represent alternative approaches to URL. For evaluation, we use numeric and pixel-based locomotion environments from the DeepMind Control Suite (DMC) (Tassa et al., 2018), along with manipulation environments in MuJoCo (Todorov et al., 2012; Schulman et al., 2016; Gupta et al., 2020). Detailed descriptions of the environments and experimental settings for each algorithm are provided in the appendix (see Appendix D).

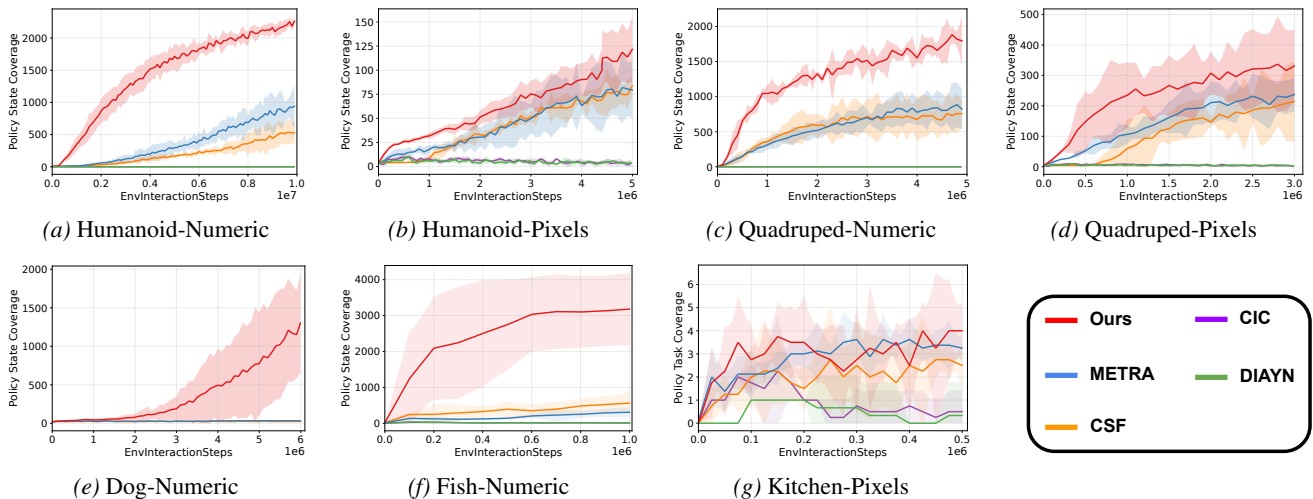

*Figure 3.* **Quantitative comparison with unsupervised skill discovery methods (4 seeds).** We measure the state/task coverage of the policies. Our algorithm scores the best coverage across all environments. Notably, our algorithm learns meaningful skills in "Dog-Numeric" and "Fish-Numeric" where other methods fail.

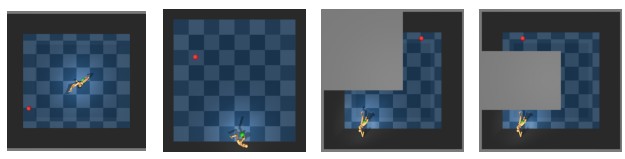

*Figure 4.* **Downstream benchmark environments for Humanoid.** Prefix F- means "Fixed" and R- means "Random". Suffix -S means "Start (green ball in image)", -G means "Goal (red ball in image)". For instance, (FS, RG) depicts the fixed initial start with a random goal in both the training and evaluation phases. Maze environments serve (RS, RG) for the training phase, and challenging (FS, FG) is given for the evaluation phase.

### 5.1. Main Results

**Skill pre-training.** In Figure 3, we evaluate the skill policy's state and task coverage. Coverage is measured by the number of unique coordinate bins (or unique tasks for Kitchen) visited across $k = 48$ evaluation episodes using randomly sampled skills. We set the bin size to 1 for the x and y dimensions in general environments and 0.01 for the x, y, and z dimensions in Fish.

In a diverse set of environments, GenDa achieves higher state coverage with substantially fewer environment interactions than prior methods. Moreover, in *Dog*—an environment with high-dimensional state (226 dims) and action (38 dims) spaces where existing algorithms often fail to learn meaningful skills—our method succeeds in learning skills. These results indicate that our relabeling approach can enhance the data efficiency in pre-training, which is also scalable to challenging tasks.

**Downstream tasks.** To investigate the generality and scalability of the pre-trained policies, we propose diverse downstream tasks.

We further evaluate on downstream tasks using the pre-trained skill policy within a hierarchical framework. Specifically, the skill policy is frozen, and a high-level task policy outputs an action $z$; the skill policy then executes the corresponding skill for a fixed number of timesteps. Each episode has a goal. If the agent reaches the given goal, it receives a +1 success reward and 0 reward for the other states. The meaning of each suffix is described in Figure 4.

(FS, RG) is the easiest setting, as it provides the global information distribution similar to that of the pre-trained policy. The (RS, RG) setting tests whether the skill policy can execute the learned skill under a shift in global information. The Maze benchmark is the most challenging task in our suite; solving it requires strong generalization and exploration capabilities to perform a given skill over a wider range of coordinates. As shown in Table 1, our GenDa achieves the highest average success rate across downstream tasks.

Crucially, GenDa achieves comparable performance to the asymptotic limit of baselines with 5× fewer environment interactions in Table 2. This superior data efficiency positions GenDa as a practical foundation for real-world robotic learning where interaction is costly.

### 5.2. GenDa ablation study

In this section, we evaluate the contribution of each component in GenDa and provide further analyses of our algorithm.

*Table 1.* **Quantitative comparison in downstream tasks (4 seeds).** High-level controllers are trained for 20k and 40k interaction steps in state-based and pixel-based environments, respectively.

| Type | Domain | Task | CSF | METRA | **Ours** |
|---|---|---|---|---|---|
| *State-based* | Humanoid | (FS, RG) | $0.35 \pm 0.34$ | $0.52 \pm 0.12$ | $\mathbf{0.83 \pm 0.08}$ |
| | | (RS, RG) | $0.08 \pm 0.06$ | $0.04 \pm 0.03$ | $\mathbf{0.68 \pm 0.13}$ |
| | | MazeE | $0.34 \pm 0.40$ | $0.19 \pm 0.27$ | $\mathbf{0.78 \pm 0.04}$ |
| | | MazeH | $0.01 \pm 0.01$ | $0.00 \pm 0.00$ | $\mathbf{0.41 \pm 0.20}$ |
| | Quadruped | (FS, RG) | $0.50 \pm 0.07$ | $\mathbf{0.81 \pm 0.05}$ | $0.78 \pm 0.08$ |
| | | (RS, RG) | $0.01 \pm 0.02$ | $0.04 \pm 0.02$ | $\mathbf{0.35 \pm 0.03}$ |
| | | MazeE | $0.14 \pm 0.28$ | $0.02 \pm 0.03$ | $\mathbf{0.33 \pm 0.33}$ |
| | | MazeH | $0.00 \pm 0.00$ | $0.00 \pm 0.00$ | $\mathbf{0.13 \pm 0.16}$ |
| | Dog | (FS, RG) | $0.01 \pm 0.01$ | $0.01 \pm 0.01$ | $\mathbf{0.55 \pm 0.10}$ |
| | | (RS, RG) | $0.00 \pm 0.00$ | $0.00 \pm 0.00$ | $\mathbf{0.53 \pm 0.08}$ |
| | | MazeE | $0.00 \pm 0.00$ | $0.00 \pm 0.00$ | $\mathbf{0.56 \pm 0.07}$ |
| | | MazeH | $0.00 \pm 0.00$ | $0.00 \pm 0.00$ | $\mathbf{0.35 \pm 0.15}$ |
| | Fish | (FS, RG) | $0.00 \pm 0.00$ | $0.00 \pm 0.00$ | $\mathbf{0.79 \pm 0.06}$ |
| *Pixel-based* | Humanoid | (FS, RG) | $0.36 \pm 0.09$ | $0.35 \pm 0.04$ | $\mathbf{0.48 \pm 0.15}$ |
| | | (RS, RG) | $0.18 \pm 0.06$ | $0.16 \pm 0.07$ | $\mathbf{0.30 \pm 0.10}$ |
| | Quadruped | (FS, RG) | $0.40 \pm 0.22$ | $0.60 \pm 0.09$ | $\mathbf{0.64 \pm 0.07}$ |
| | | (RS, RG) | $0.18 \pm 0.08$ | $0.30 \pm 0.11$ | $\mathbf{0.40 \pm 0.11}$ |

*Table 2.* **Downstream performance of GenDa in Humanoid-Numeric downstream tasks (4 seeds).** We compare the performance between two pre-trained skill policies of GenDa with different numbers of interaction steps.

| Env | Ours(2M) | Ours(10M) |
|---|---|---|
| Humanoid-Numeric-(FS, RG) | $0.73 \pm 0.14$ | $\mathbf{0.83 \pm 0.08}$ |
| Humanoid-Numeric-(RS, RG) | $\mathbf{0.77 \pm 0.16}$ | $0.68 \pm 0.13$ |
| Humanoid-Numeric-MazeE | $0.60 \pm 0.32$ | $\mathbf{0.78 \pm 0.04}$ |
| Humanoid-Numeric-MazeH | $0.14 \pm 0.17$ | $\mathbf{0.41 \pm 0.20}$ |

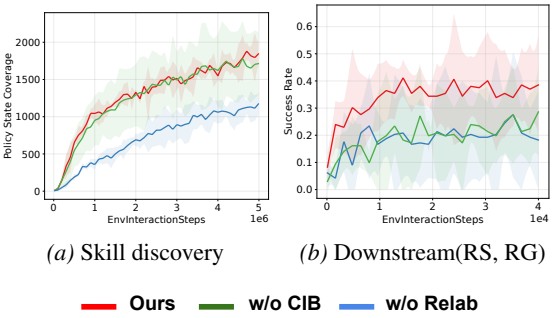

*(a)* Skill discovery  *(b)* Downstream(RS, RG)

**— Ours   — w/o CIB   — w/o Relab**

*Figure 5.* **Component analysis in Quadruped-Numeric (4 seeds).**

**Component-wise Evaluation.** We conduct ablation studies to quantify the contribution of each module in GenDa. Figure 5 demonstrates that each component addresses a distinct challenge in the URL pipeline. The W/O RELAB variant fails to acquire meaningful skills efficiently during the pre-training phase. These empirical results confirm that our skill relabeling is essential to address off-policy non-stationarity and maintain robust **data efficiency**. In contrast, W/O CIB achieves high state coverage during pre-training but struggles in the (RS, RG) downstream task. This discrepancy highlights that while standard objectives may increase coverage, they often lead to context-dependent policies. The CIB is therefore essential for disentangling the policy from the global context to ensure **generalizability**. Collectively, these results confirm that our components are complementary: Relabeling enables efficient learning, while CIB ensures the learned skills are robust and scalable.

$\beta$ **Analysis.** We conducted additional skill pre-training experiments across diverse environments using various values of $\beta$ to verify its robustness. The results in Table 3 highlight

that while performance varies depending on $\beta$, the model is not hyper-sensitive to the point where slight variations trigger an objective collapse in either numeric or pixel domains. This provides evidence that our proposed objective, driven by skill relabeling and the uniformity regularizer, can be deployed across varied environments without requiring exhaustive per-environment tuning.

### 5.3. GenDa analysis

**Update-to-Data (UTD) Analysis.** Figure 6 serves as empirical evidence for the non-stationary hypothesis. While baselines suffer from instability at high UTD ratios due to semantic drift, GenDa effectively utilizes frequent updates. This confirms that our skill relabeling turns stale $z$ of the replay buffer into a consistent source of supervision, enhancing the potential of off-policy learning. This result confirms our theoretical hypothesis in Appendix B.2: Variance of se-

*Table 3.* **Performance across different environments varying $\beta$ parameter (3 seeds).**

| $\beta$ | Humanoid-Numeric | Quadruped-Numeric | Quadruped-Pixel | Humanoid-Pixel |
|---|---|---|---|---|
| **0.1** | $2094 \pm 89$ | $1740 \pm 223$ | $286 \pm 85$ | $127 \pm 72$ |
| **0.5** | $2236 \pm 103$ | $1597 \pm 66$ | - | - |
| **1.0 (ours)** | $2192 \pm 112$ | $1794 \pm 220$ | $331 \pm 75$ | $124 \pm 37$ |
| **2.0** | $2213 \pm 192$ | $1531 \pm 409$ | - | - |
| **4.0** | $2595 \pm 386$ | $1783 \pm 63$ | $285 \pm 77$ | $123 \pm 30$ |

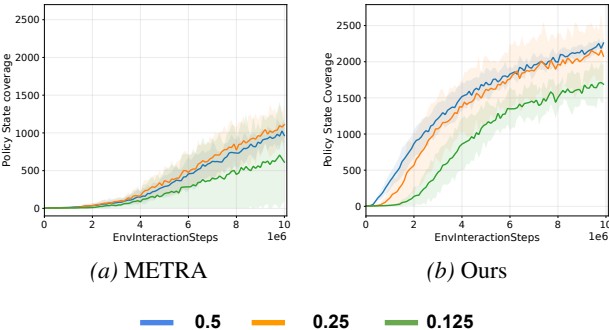

| | *(a)* METRA | *(b)* Ours |
|---|---|---|

| **— 0.5** | **— 0.25** | **— 0.125** |
|---|---|---|

*Figure 6.* **Update-to-Data (UTD) ratio test in Humanoid-Numeric (4 seeds).** $0.125$ is a commonly used setting in prior work.

mantic drift in replay buffer via relabeling is the key factor enabling data-efficient off-policy learning.

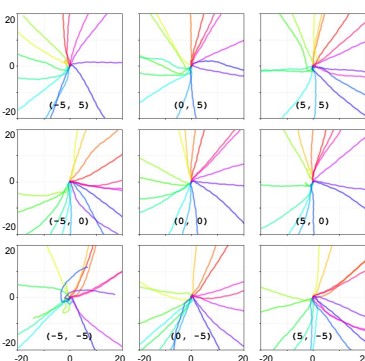

*Figure 7.* **Trajectories for various offsets in Humanoid-Numeric Environment.** Same color means the same skill $z$ is given to our skill policy. An offset (a, b) has the same meaning as in Figure 2.

**Global information robustness.** To assess our research question 2, we investigate how robust GenDa is to the generalization issue induced by the global context discussed in Section 3.2. The qualitative results in Figure 7 show GenDa has better generalizability by using CIB that encourages the skill policy to rely on the given skill and its semantic meaning rather than global context.

*Table 4.* **Comparison of performance with unseen background conditions in Quadruped-Pixels (4 seeds).** 'Unicolor' and 'Gradation' denote evaluation settings using four distinct solid colors and gradient patterns, respectively.

| | Ours | Ours(w/o CIB) |
|---|---|---|
| **Original** | $331 \pm 75$ | $\mathbf{335 \pm 103}$ |
| **Unicolor** | $\mathbf{268 \pm 25 (81\%)}$ | $137 \pm 99 (41\%)$ |
| **Gradation** | $\mathbf{206 \pm 88 (62\%)}$ | $78 \pm 47 (23\%)$ |

**Complementary encoder analysis.** To investigate whether our CIB has a complementary relationship with $\phi$, we decode the remaining information from each representation. Figure 8 shows that $\phi$ mainly retains background color while losing recognizable proprioception, whereas CIB retains proprioception while losing background color.

We also tested the skill policy's robustness by applying unseen visual shifts to the background in pixel-based environment. Consequently, the results in Table 4 demonstrate that the CIB provides relative robustness compared to the W/O CIB in pixel environments, but its effectiveness is degraded under the visual shifts. We abstained from using any image augmentation techniques during visual training to ensure a fair comparison with the baselines. This highlights a highly promising direction: because the CIB module structurally disentangles redundant context, incorporating standard visual generalization techniques (Yarats et al., 2022) during CIB training would seamlessly bridge this gap and yield a model highly robust to visual characteristics.

## 6. Related Work

**Unsupervised Skill Discovery** Unsupervised reinforcement learning (URL) aims to learn skill representations, policies, or dynamics models through interaction with the environment without any external task reward (Eysenbach et al., 2019; Warde-Farley et al., 2019; Campos et al., 2020; Hartikainen et al., 2020), and then reuse them for downstream learning (Zhao et al., 2022; Laskin et al., 2022; Yang et al., 2024). Prior unsupervised skill discovery methods train a skill-conditioned policy from states or trajectories by maximizing mutual information objectives (Eysenbach et al., 2019; Laskin et al., 2022; Strouse et al., 2022). They en-

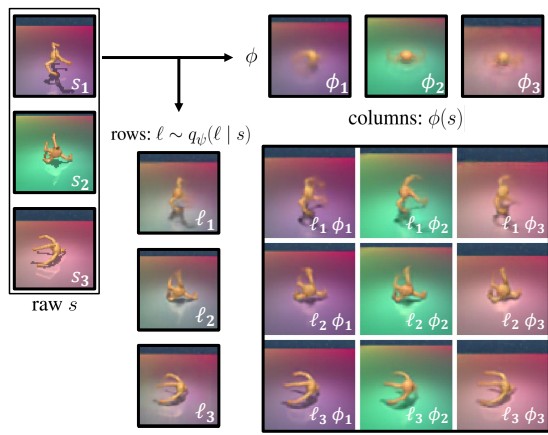

*Figure 8.* **Remaining information of CIB encoder and $\phi$.** For the bottom-right $3 \times 3$ decoded images, we use the CIB decoder $p(s \mid \ell, \phi(s))$. For the decoded images of $\ell_i$ and $\phi_i$, we train independent decoders that do not affect the base encoder.

able the learning of diverse and distinguishable behaviors even without external rewards. Some approaches further leverage dynamics models to capture mutual information at the state-transition level (Sharma et al., 2020; Mendonca et al., 2021). However, these methods can prioritize discriminability over how far behaviors spread in the environment. As a result, in high-dimensional state spaces, they have been reported to discover redundant or locally confined skills (Park et al., 2022; 2024; Zheng et al., 2025). To mitigate this, more recent advances in URL leverage explicit state-space metrics to discover skills with Wasserstein dependency measure (Ozair et al., 2019; He et al., 2022) and InfoNCE (van den Oord et al., 2018; Ma & Collins, 2018; Henaff, 2020; Myers et al., 2024) that transfer better to downstream tasks (Park et al., 2024; Zheng et al., 2025). Despite the promise of URL, prior methods face two limitations: (1) data efficiency and (2) policy generalization.

**Disentanglement in unsupervised skill discovery.** Some prior work has explored inducing disentanglement by explicitly structuring the latent space via factorization (Hu et al., 2024; Wang et al., 2024). Another line of work proposes masking the policy input to encourage structured skill usage (Park et al., 2026). However, these methods tend to rely on assumptions such as factorized factors or periodic structure. In contrast, our complementary variational encoder (CIB) improves skill reusability by modulating the skill policy's input conditioning, without a hand-crafted structure.

## 7. Conclusion

### 7.1. Implication for Unsupervised Skill Learning

We identify two fundamental challenges in off-policy unsupervised reinforcement learning that limit the scalability of

skill discovery. Our skill relabeling mitigates the semantic drift in prior URL methods together with a compatible regularization term. It significantly improves the data efficiency in the pre-training phase. We also point out a failure mode of the skill policy in downstream tasks, which is caused by global contextual information. With our drop-in CIB module, the generalizability of the skill policy is improved in various downstream tasks.

### 7.2. Open Challenges and Future Directions

**Skill relabeling** Our skill relabeling still inherits a *one-skill-per-episode* assumption that can be violated when a single episode contains heterogeneous behaviors (e.g., exploratory "moving around" segments interleaved with skill-consistent transitions). Our relabeling target is derived from the telescoping sum; thus, it can be generalized to any arbitrary time segment. So, our skill relabeling framework is structurally and mathematically equipped to handle intra-episode skill changes. We will explore extending relabeling to assigning time-varying skill labels, which may better capture abrupt changes in intent, such as sharp direction switches or rapid reactive adjustments.

**Generalization** CIB is designed to disentangle the global context from the skill policy; its effectiveness depends on the learned metric-aware latent function $\phi$; if one wishes to apply the idea to non-metric-aware skill discovery algorithms, this dependency can limit applicability. We consider it a feasible future direction to use *dynamics-aware* objectives rather than a particular embedding metric.

**Manipulation task** Our metric-aware objective, which maximizes temporal state distances, inherently encourages large-scale movements. This objective hinders the acquisition of the fine-grained, small-scale behaviors required for precise manipulation. Addressing this will require incorporating magnitude-conditioned skill mechanisms, such as PSD (Park et al., 2026), to balance large explorations with delicate object manipulation.

## Impact Statement

This paper presents work whose goal is to advance the field of Machine Learning. There are many potential societal consequences of our work, none of which we feel must be specifically highlighted here.

## Acknowledgements

This work was partly supported by the National Research Foundation of Korea (NRF) (No. RS-2024-00348376), by the Institute of Information & Communications Technology Planning & Evaluation (IITP) (No. RS-2024-00438686, No.

RS-2022-II221045, No. RS-2025-02218768, No. RS-2019-II190421) grant funded by the Korea government (MSIT), and by the Industrial Technology Innovation Program (No. RS-2025-25448266) funded by the Ministry of Trade, Industry & Energy (MOTIE) and Korea Evaluation Institute of Industrial Technology (KEIT).

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

# A. Relabel-and-Optimize Procedure

**Setting.** Let $B$ be a fixed replay buffer containing either transitions $(s, s', z)$ or trajectories $\tau$ with endpoints $(s_0(\tau), s_T(\tau))$. Let $V \subseteq \mathcal{S}$ denote the finite set of states that appear in the dataset (tabular setting), and restrict the optimization variable to

$$\phi : V \to \mathbb{R}^d.$$

Hence $\phi$ can be identified with a vector in the finite-dimensional space $\mathbb{R}^{d|V|}$.

We encode the Lipschitz constraints via an undirected graph $G = (V, E)$:

$$\{u, v\} \in E \iff \text{a transition } u \to v \text{ or } v \to u \text{ is observed in the dataset.}$$

We fix the gauge by choosing an anchor state $s_0 \in V$ and enforcing $\phi(s_0) = 0$.

**Assumption A.1** (Connected relevant support)**.** All states that appear in the $\phi$-step objective lie in a single connected component of the original graph $G$. Without loss of generality, we restrict $V$ and $E$ to the induced subgraph on this component and choose an anchor $s_0 \in V$. Hence $G = (V, E)$ is connected.

**Relabeling map.** For $\varepsilon > 0$, define

$$g(\phi; a, b) = \frac{\phi(b) - \phi(a)}{\max(\|\phi(b) - \phi(a)\|, \varepsilon)}.$$

Then $g$ is well-defined for all $(a, b)$ and continuous, with $\|g(\phi; a, b)\| \leq 1$. While $g$ may not be differentiable on the boundary $\|\phi(b) - \phi(a)\| = \varepsilon$, differentiability is not required for the existence results below.

**Relabel-and-optimize procedure.** Given the current $\phi$,

1. **(z-step; relabel)** For each trajectory $\tau$, set

$$z_\tau \leftarrow g(\phi; s_0(\tau), s_T(\tau)).$$

2. **($\phi$-step; optimize)** Treat $\{z_\tau\}$ as constants during the $\phi$-update (i.e., apply a stop-gradient through $z_\tau$) and solve

$$\max_\phi \quad \sum_\tau \big\langle \phi\big(s_T(\tau)\big) - \phi\big(s_0(\tau)\big), \; z_\tau \big\rangle \tag{11}$$

$$\text{s.t.} \quad \|\phi(u) - \phi(v)\| \leq 1 \quad \forall\{u, v\} \in E, \tag{12}$$

$$\phi(s_0) = 0. \tag{13}$$

We emphasize that we do not claim global monotonic improvement or convergence of the full iteration without additional structure; instead, we establish that each step is well-defined and that the $\phi$-step admits an optimizer at every iteration.

*Proof Sketch.* **Feasibility.** The feasible set is non-empty since $\phi(s) \equiv 0$ for all $s \in V$ satisfies (12) and (13).

**Boundedness.** Fix any $s \in C(s_0)$. Let $s_0 = v_0, v_1, \ldots, v_L = s$ be a shortest path in $G$. By the triangle inequality and the Lipschitz constraints,

$$\|\phi(s)\| = \|\phi(v_L) - \phi(v_0)\| \leq \sum_{i=1}^L \|\phi(v_i) - \phi(v_{i-1})\| \leq \sum_{i=1}^L 1 = L = \text{dist}_G(s_0, s).$$

Thus each coordinate block $\phi(s)$ is bounded over the feasible set.

**Compactness.** Since $V$ is finite, $\phi$ lies in a finite-dimensional Euclidean space. Constraints (12)–(13) define a closed set (inverse images of continuous functions), and the feasible set is bounded by the previous step; hence it is compact (Heine–Borel).

**Existence.** The objective (11) is linear (hence continuous) in $\phi$ and the feasible set is compact, so an optimizer exists by the Weierstrass theorem.

**Global optimality.** The objective is linear (thus concave) and the constraints are convex, so this is a convex optimization problem. Therefore any optimizer is globally optimal. □

**Proposition A.2** (Existence and global optimality of the $\phi$-step solution)**.** *Under Assumption A.1 (tabular setting), for any fixed relabeled signals $\{z_\tau\}$, the $\phi$-step problem (11)–(13) admits at least one optimal solution $\phi^\star$. Moreover, the problem is a convex optimization problem, hence any optimizer is globally optimal.*

**Corollary A.3** (Per-iteration existence in the idealized tabular setting)**.** *For $\varepsilon > 0$, the relabeling map is well-defined. At every iteration, for the fixed relabeled signals produced by the $z$-step, the idealized tabular $\phi$-subproblem admits at least one globally optimal solution.*

## B. Relabeling under Non-Stationary Label Noise

At a fixed current update, we view replay-buffer labels as an endpoint-conditioned mixture of labels collected at different training times; the resulting conditional covariance represents variability induced by non-stationary skill semantics.

### B.1. Non-Stationary with a fixed z label.

**Noise model**    For each trajectory $\tau$ with endpoints $(s_0(\tau), s_T(\tau))$, suppose the original label $z_\tau$ is noisy:

$$\mathbb{E}[z_\tau \mid s_0, s_T] = \mu(s_0, s_T), \qquad \mathrm{Cov}(z_\tau \mid s_0, s_T) = \Sigma(s_0, s_T) \succeq 0.$$

In non-stationary settings, the conditional covariance matrix $\Sigma(s_0, s_T)$ may vary across endpoint pairs, and typically $\Sigma(s_0, s_T) \neq 0$ on parts of the support.

**Alignment scalar and conditional variability.**    For a fixed $\phi$, define the per-trajectory alignment contribution

$$Y_\tau(\phi) \;=\; \big\langle \phi\big(s_T(\tau)\big) - \phi\big(s_0(\tau)\big), \; z_\tau \big\rangle.$$

The conditional variance is

$$\mathrm{Var}\big(Y_\tau(\phi) \mid s_0, s_T, \phi\big) = \Delta_\phi^\mathsf{T} \Sigma(s_0, s_T) \Delta_\phi \geq 0,$$

where $\Delta_\phi(s_0, s_T) := \phi(s_T) - \phi(s_0)$. It is strictly positive if and only if

$$\Delta_\phi^\mathsf{T} \Sigma(s_0, s_T) \Delta_\phi > 0,$$

i.e., when the label-noise covariance has nonzero variance along the current alignment direction.

### B.2. $z$ relabeling and label noise.

**Relabeling removes conditional label-noise variability (from the $\phi$-step perspective).**    Define the relabeled signal deterministically via

$$\tilde{z}_\tau \;=\; g(\phi; \, s_0(\tau), s_T(\tau)),$$

and in the $\phi$-step treat $\tilde{z}_\tau$ as a constant (stop-gradient). Then the relabeled alignment term

$$\tilde{Y}_\tau(\phi) \;=\; \big\langle \phi\big(s_T(\tau)\big) - \phi\big(s_0(\tau)\big), \; \tilde{z}_\tau \big\rangle$$

satisfies

$$\mathrm{Var}\big(\tilde{Y}_\tau(\phi) \mid s_0, s_T, \phi\big) = 0.$$

Thus, from the perspective of the $\phi$-step update, conditional variability due to label noise is removed.

**Implication: potential reduction in stochastic-gradient variance (with possible bias).**    Because the per-sample gradient of (11) is affine (here linear) in the label $z$, the law of total variance suggests decomposing stochastic-gradient variance into (i) a component attributable to conditional label noise and (ii) a component attributable to sampling. When using the original noisy labels, the former component is nonzero whenever the corresponding quadratic form induced by $\Sigma(s_0, s_T)$ is positive. When using relabeling (and treating $\tilde{z}_\tau$ as constant in the $\phi$-step), this conditional label-noise component can drop to zero, potentially reducing gradient variance and improving update stability in non-stationary regimes.

However, since $\tilde{z}_\tau$ depends on $\phi$, relabeling can introduce bias relative to updates driven by the original noisy labels. Hence relabeling reflects a bias–variance trade-off: it may improve stability when label noise dominates, provided the induced bias remains controlled.

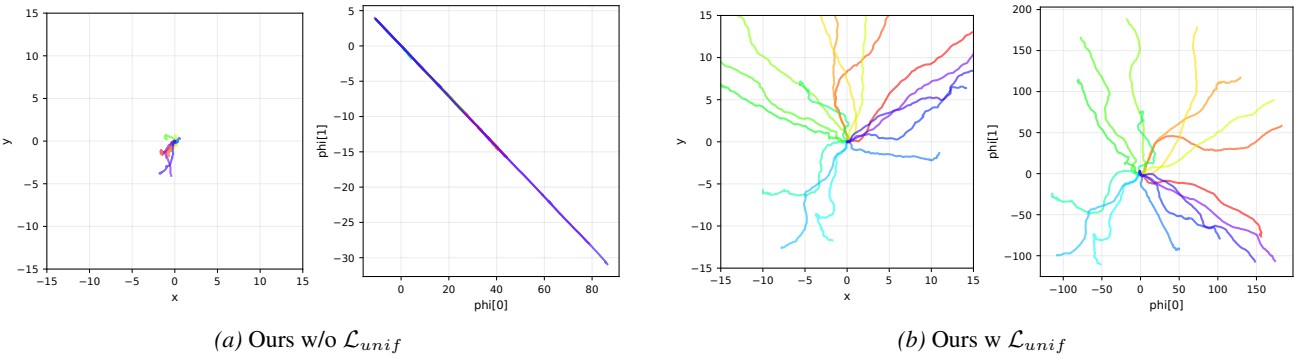

*(a)* Ours w/o $\mathcal{L}_{unif}$        *(b)* Ours w $\mathcal{L}_{unif}$

Left of each pair: $(\tau_z)$, Right of each pair: $(\phi(\tau_z))$

*Figure 9.* **Trajectory $\tau_z$ and its latent $\phi(\tau_z)$ in Humanoid-Numeric at 1M timesteps.** $\tau_z$ is a $z$-conditioned trajectory, and the same color indicates the same skill $z$ is given. Our uniformity term can encourage the latent space to have diverse directions spread in the early step of learning.

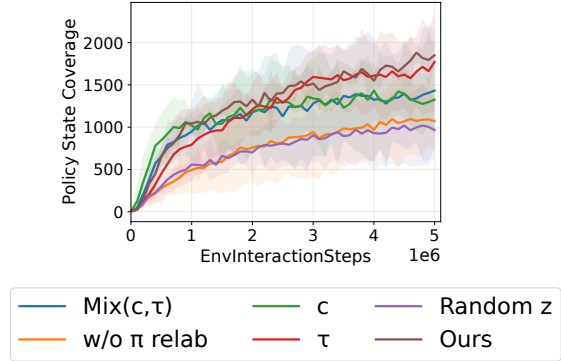

*Figure 10.* **Skill policy intrinsic reward z in Quadruped-Numeric(4 seeds).** $z_{roll}$: Using fixed z only, C: c-step relabel, $\tau$: $0 \to T$ relabel, RANDOM Z: newly sampled random z, OURS: Mix($c, \tau, z_{roll}$)

## C. Policy Relabeling Analysis.

To investigate the proper $z$ conditions for the intrinsic reward function, we evaluate multiple relabeling strategies that can be used for selecting target $z$ and report their comparative results in Figure 10.

For Mix(c,$\tau$), we use a 0.5:0.5 ratio. For Ours, we compute the probability of $z_{roll}$ using a count-based estimator with respect to the $z_{relab}$ directions of the episodes stored in the replay buffer. This estimator is updated like an exponential moving average (EMA) each time a $z_{roll}$ is sampled from the buffer for learning. If the resulting probability is lower than threshold $\epsilon_z = 0.4$, we keep the corresponding $z_{roll}$; otherwise, we relabel it using $\text{Mix}(c, \tau)$. We also test multiple settings for the c-$\tau$ mixture ratio and the threshold $\epsilon_z$ in Table 5, but observe only marginal differences in performance.

## D. Implementation details

We implement our algorithm and baselines with JAX for faster training. Our code is based on the official PyTorch implementation and paper of each baseline. For stable learning, we use the LayerNorm in the critic networks of both our method and the baselines. For CIC and DIAYN, we use the hyperparameters used in the official CSF implementation. For METRA, CSF and Ours, we use same hyperparameters except high-level controllers in downstream tasks that are represented in Table 8.

### D.1. Environment settings

**Downstream tasks.** To evaluate performance on downstream tasks, we utilize 6 domains, each containing 1 to 4 specific tasks. High-level controllers $\pi^h(z|s, s^{task})$ selects a skill every K=25 environment steps, where $s^{task}$ denotes the task

*Table 5.* **State coverage corresponding to different mixture ratios and thresholds in Quadruped-Numeric** (3 seeds).

| $c - \tau$ **mixture ratio** | **State Coverage** |
|---|---|
| **0.5:0.5 (original)** | $1851 \pm 123$ |
| **0.25:0.75** | $1820 \pm 241$ |
| **0.75:0.25** | $1842 \pm 229$ |

| **Threshold for** $z_{roll}$ | **State Coverage** |
|---|---|
| **0.4 (original)** | $1851 \pm 123$ |
| **0.8** | $1817 \pm 245$ |
| **0.95** | $1678 \pm 318$ |

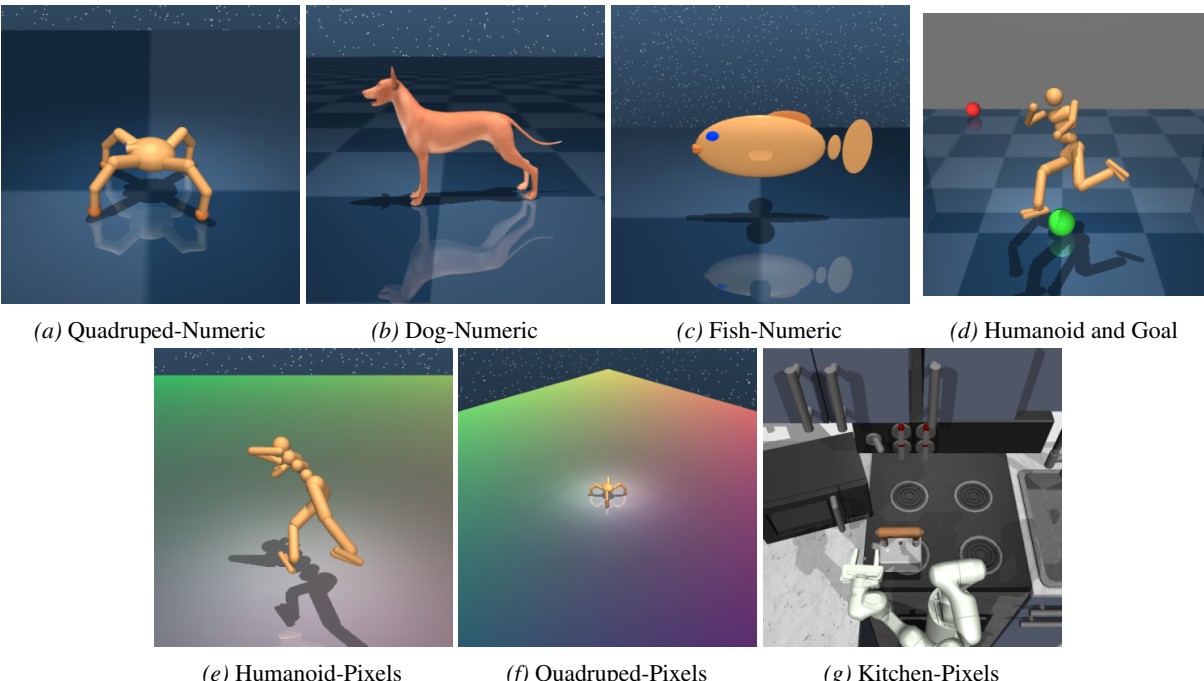

*(a)* Quadruped-Numeric     *(b)* Dog-Numeric     *(c)* Fish-Numeric     *(d)* Humanoid and Goal

*(e)* Humanoid-Pixels     *(f)* Quadruped-Pixels     *(g)* Kitchen-Pixels

*Figure 11.* **Benchmark environments for various domains.**

information (like desired goal).

**Maze tasks.** The agent navigates a $7 \times 7$ maze to reach a goal. During training, both the agent and the goal are initialized at random feasible positions, with a minimum separation distance of 3. During evaluation, the agent starts from a fixed location $(1, 0)$. The goal is placed at $(5, 6)$ for *Easy* and at $(1, 6)$ for *Hard*. For the Quadruped agent, we scale the maze dimensions by a factor of 1.5 to account for its larger body size. We set the episode horizon to 400 environment steps for *Easy* and 500 steps for *Hard*.

**(RS, RG) tasks (Quadruped, Humanoid, Dog-Numeric).** Both the agent and the goal are initialized uniformly at random in $[-3.5, 3.5]^2$, with a minimum separation distance of 3. As in the Maze tasks, we increase the environment scale by a factor of 1.5 for the Quadruped agent.

**(FS, RG) tasks.** The agent starts from the origin $(0, 0)$, while the goal is sampled randomly. Specifically, the goal is sampled from $[-3.5, 3.5]^2$ for Humanoid-Numeric and from $[-5, 5]^2$ for Humanoid and Quadruped-Pixels. In the Fish environment, the agent starts from a fixed position with a random orientation (quaternion), and the goal is placed 0.5 units ahead along the agent's forward direction. As in the Maze tasks, we increase the environment scale by a factor of 1.5 for the Quadruped agent.

For (FS, RG) and (RS, RG), we set the episode horizon to 200 environment steps for *Quadruped (Num/Pix)*, *Humanoid*

*Table 6.* **Environment parameters.**

| ENV | $s$ DIM | $a$ DIM | FRAME STACKS(ACTION REPEATS) | EPS. LENGTH |
|---|---|---|---|---|
| HUMANOID-NUMERIC | 70 | 21 | 1(2) | 200 |
| QUADRUPED-NUMERIC | 81 | 12 | 1(2) | 200 |
| DOG-NUMERIC | 226 | 38 | 1(2) | 200 |
| FISH-NUMERIC | 27 | 5 | 1(2) | 200 |
| HUMANOID-PIXELS | $64\times64\times3$ | 21 | 3(2) | 200 |
| QUADRUPED-PIXELS | $64\times64\times3$ | 12 | 3(2) | 200 |
| KITCHEN-PIXELS | $64\times64\times3$ | 7 | 3(1) | 50 |

*(Num/Pix)*, and *Fish*, and to 400 steps for *Dog-Numeric*.

*Table 7.* **Hyperparameters for unsupervised skill discovery methods.**

| Hyperparameter | Value |
|---|---|
| Learning rate | 0.0001 |
| Optimizer | Adam |
| # episodes per epoch | 2(-Numeric), 8(Kitchen), 4(Humanoid, Quadruped-Pixels) |
| # gradient steps per epoch | 200 (others), 50 (Kitchen) |
| Minibatch size | 256 |
| Discount factor | 0.99 |
| Replay buffer size | $10^6$(-Numeric), $3 \times 10^5$(-Pixels) |
| Encoder | CNN |
| # hidden layers | 2 |
| # hidden units per layer | 1024 |
| Target network smoothing coefficient | 0.995 |
| Entropy coefficient | 0.01(Kitchen), auto-adjust(others) |
| Ours $z$ dim | 2 (Humanoid-, Quadruped-, Dog-), 3 (Fish-), 10 (Kitchen) |
| METRA, CSF $z$ dim | 2 (Humanoid-, Quadruped-, Dog-), 3 (Fish-), 24-discrete (Kitchen) |
| DIAYN $z$ dim | 50-discrete |
| CIC $z$ dim | 64 |
| Ours, METRA $\epsilon_\lambda$ | $10^{-3}$ |
| Ours, METRA initial $\lambda$ | 30 |
| Ours $\ell$ dim | 256(-Numeric), 2048(-Pixels) |
| Ours $\beta$ | 1.0 |
| Ours $\eta$ | 0.1 |

*Table 8.* **Hyperparameters for SAC in high-level controllers.**

| Hyperparameter | Value |
| --- | --- |
| Actor learning rate | 0.0003 |
| Critic learning rate | 0.0003 |
| Entropy coefficient learning rate | 0.001 |
| Action frequency | 25 |
| Optimizer | Adam |
| # episodes per epoch | 8 |
| # gradient steps per epoch | 200 (others), 50 (Kitchen) |
| Minibatch size | 256 |
| Discount factor | 0.99 |
| Replay buffer size | $10^6$(-Numeric), $3 \times 10^5$(-Pixels) |
| Encoder | MLP for state inputs; CNN for pixel inputs |
| # hidden layers | 2 |
| # hidden units per layer | 1024 |
| Target network smoothing coefficient | 0.995 |
| Entropy coefficient | 0.01 (Kitchen), auto-adjust (others) |

