# OpenReview forum: "Learning Generalizable Skill Policy with Data-Efficient Unsupervised RL"
_ICML.cc/2026/Conference — ICML 2026 regular_

### Official Review · Reviewer_a9e4 · 2026-03-04

**Soundness:** 2
**Presentation:** 3
**Significance:** 3
**Originality:** 3
**Overall Recommendation:** 4
**Confidence:** 4

**Summary:**

This paper propose GenDa (Generalizable Data-efficient Agent), addressing two challenges in the realm of off-policy unsupervised reinforcement learning (URL). The first problem is the semantic drift due to modern URL methods often rely on off-policy algorithms that reuse past experiences stored in a monotonic replay buffer. In the later stage of training, trajectories sampled by the same skill can change as the skill can evolve. The second problem is the poor generalization ability from global context overfitting. To address these challenges, authors 1) formalize and analyze two failure models, 2) introduce skill relabeling to mitigate semantic drift, 3) propose a regularizer to ensure skill diversity under the Mutual Information (MI), and 4) propose a CIB (Complementary Information Bottleneck) that learns an embedding that prevents acquiring excessive global contextual information leads to overfitting. Experiments show GenDa 1) outperforms SOTA URL algorithms in exploration (i.e., state converages) in various control problems, and 2) demonstrates robustness in tranfering to downstream tasks.

**Compliance With Llm Reviewing Policy:**

Affirmed.

**Final Justification:**

Most of my concerns has been addressed, and I am happy to raise my score to a positive score.

**Key Questions For Authors:**

1. EMA seems to be a great way of soothing semantic drift, can you elaborate more on this, and explain if there are other methods available, and why EMA is selected?
2. How necessary and effective of the uniformity regularizer?
3. Can you summarize why CIB can prevent overfitting?
4. It is possible or necessary to apply GenDa to similar areas, like on-policy RL?
5. This is minor. Why "data-efficient" is emphasized in the title, but I cannot find a single dedicated section or sentence to explain why this method is "data-efficient"? Actually, "data-efficient" is not your motivation nor selling point.

**Limitations:**

Yes

**Strengths And Weaknesses:**

- Strengths
  - Using exponential moving average (EMA) to soothes semantic drift is novel and intuitive.
  - Taking care of representation collapse is good. I am curious if authors can deliver any ablation study on that.
  - The idea of relabling for skill representaiton / policy learning is valid. I also see the potential of applying this method to other similar areas.
- Weaknesses
  - In section 3, figure 1's caption states, "same skill z can be associated with difference trajectories depending on the training stage ...". This is an important and very valuable claim, but how can you justify this via figure 1? Should be explained better. Also, figure 2 is confused, what are these curves and the "offset"? Readers may not have prior knowledge on Humanoid-Numeric Environment, please describe and explain briefly.
  - The presentation of algorithm 1 should be improved. This algorithm itself is the distillation of the entire section 4, and authors are expected to formalize and explain it in a better way.
  - This is minor, line 113 "same skill z". This skill "z" can be confused with previous latent skill vectors.
  - This is minor, other typos and formatting problems should be fixed.
- Soundness: 2
  - The theoretical foundation of the proposed method appears valid, but the paper currently lacks sufficient empirical evidence and ablation studies to fully justify its claims regarding skill-trajectory association and representation collapse.
- Presentation: 2
  - Some parts of the paper writing could be improved (see comments).
- Significance: 3
  - Generalization is always important, and the proposed solutions for skill representation offer valuable insights that could benefit similar areas in reinforcement learning.
- Originality: 3
  - The introduction of an exponential moving average (EMA) to mitigate semantic drift is novel and intuitive.

---

> ### Author Rebuttal · Authors · 2026-03-25
>
> We sincerely thank the reviewer for recognizing the novelty of our approach and the validity of our theoretical foundation. We address your questions and concerns below.
>
> **Clarification of Figures 1 and 2:**
> We agree with your concern that these two figures are important to deliver the emperical evidence of defined problems.
> We will revise them in the updated manuscript.
> * **Figure 1:** Plots physical trajectories collected by the exact same skill vector at different training stages. In standard off-policy URL, these varying trajectories share the exact same label $z$ for representation learning in the replay buffer. This creates a one-to-many mapping that destabilizes representation learning, leading the semantic drift.
> * **Figure 2:** Demonstrates overfitting to xy-coordinates (global context). Offset(a,b) denotes that xy-position of init-state is shifted to (a,b) coordinates. The same colored curves represent physical trajectories generated by the same skill. The agent fails to reproduce the same coverage for the same skill when given offset is varied.
>
> **Presentation:**
> We will revise Algorithm 1 to formalize equations more clearly, accurately distilling the two-phase relabeling and optimization steps described in Section 4. Also, we will thoroughly revise typos and formatting.
>
> **Clarification on EMA and Semantic Drift:**
>
> We will revise the **Section 4.1.1** in the updated manuscript to prevent the confusion, which may have led to the misunderstanding that the EMA itself resolves semantic drift.
>
> To clarify, the core mechanism that mitigates the high-variance noise caused by semantic drift is our skill relabeling for representation learning. Unlike prior methods that naively reuse the fixed $z$ from the replay buffer, we decouple the process into a self-predictive skill relabeling phase and a subsequent optimization phase. It is this relabeling process that directly addresses and resolves the semantic drift.
>
> We adopted the EMA for a practical tool to ensure training stability and enhance performance. Using an EMA for the target network is a proven, standard technique in self-predictive representation learning (e.g., [1]).
>
> **Necessity and effectiveness of the Uniformity Regularizer:**
> The uniformity regularizer is absolutely critical. Without it, the representation easily collapses to trivially satisfy the objective, as depicted in **Figure 9(a) in the Appendix**.
>
> |w/o regularizer|State Coverage|
> |:---|:---|
> |**Quadruped-Numeric**|114±34 (-94%)|
> |**Humanoid-Numeric**|370±164 (-83%)|
>
> The contrastive uniformity regularizer explicitly prevents the collapse by forcing relabeled skill vectors to spread evenly across the hypersphere, successfully maintaining diversity. We will add these results to the ablation section.
>
> **Why CIB prevents overfitting:**
> Skill policy $\pi(a \mid s, z)$ learns to maximize an intrinsic reward that references $\phi(s)$. Driven by this objective, they often fall into an optimization shortcut by exploiting redundant, reward-correlated information in the raw state $s$ that is already represented in $z$. This entangles the policy, making it interpret $z$ strictly within a specific state context. CIB prevents this by learning an embedding $l$ explicitly trained to retain only state information *not* already captured by $\phi$. Filtering out this redundant context forces the policy to rely on the robust semantic meaning of $z$.
>
> **GenDa application to other areas:**
> We thank the reviewer for recognizing the broader potential of our research.
> * **On-policy URL:** These methods discard data after updates, avoiding the "fixed-$z$" problem. While relabeling isn't directly necessary, URL is inherently data-hungry, making the data-efficient advantages of our off-policy approach highly significant.
> * **Offline URL:** Our self-predictive relabeling learns representations even when passively labeled $z$ are completely absent from the data. This makes it highly effective for Offline URL using task-agnostic, reward-free, and even in skill label-free data.
> * **Goal-Conditioned RL (GCRL) & Hierarchical RL (HRL):** CIB is highly applicable to GCRL, where the goal $g$ is a subset of $s$, mirroring the structural entanglement CIB resolves. Similarly, in HRL, subgoals from higher-level policies act as global context for low-level policy. CIB can serve as a entanglement regularizer in both architectures to prevent lower-level policies from overfitting to global contexts.
>
> **Data-efficient keyword:**
> In our paper, data-efficient refers to sample efficiency in terms of environment interactions. As shown in Table 2 and Figure 3, GenDa achieves comparable performance with up to 1/5 interactions of the baselines. We will revise the manuscript to clearly define data-efficient and better connect it to our empirical results.
>
> **Reference**
>
> [1] Schwarzer, Max et al. “Data-Efficient Reinforcement Learning with Self-Predictive Representations.” International Conference on Learning Representations (2020)

---

> > ### Author Rebuttal · Reviewer_a9e4 · 2026-04-04
> >
> > Thanks for your response. Most of my concerns has been addressed, and I am happy to raise score once the revised manuscript is ready.

---

> > > ### Author Response · Authors · 2026-04-04
> > >
> > > We sincerely thank the reviewer for the positive feedback and the willingness to raise the score.
> > >
> > > While we would be more than happy to share the entire revised manuscript with you, the current rebuttal platform does not support full document uploads. Therefore, to assure you that your feedback has been fully incorporated, we share the revisions we have made to the manuscript addressing your main concerns below:
> > >
> > > **1. Revised Caption for Figure 1**
> > > > **Semantic drift in prior algorithms.** This figure shows trajectories collected by the skill policy of [Park et al., 2024] using the same skill vector z, illustrating that different behavioral trajectories can be observed for the same z during training. In standard off-policy URL, these trajectories are all stored in the replay buffer and repeatedly reused for representation learning. As a result, a one-to-many mapping between z and trajectories emerges, which destabilizes representation learning and leads to semantic drift.
> > >
> > > **2. Revised Caption for Figure 2**
> > > > **Overfitting to xy-coordinates (global context).** An offset (a,b) indicates that the agent’s initial xy position is set to (a,b) at evaluation time. During training, the agent always starts from (0,0), so this setup evaluates whether a given skill z produces consistent behavior under shifted initial conditions. Curves of the same color represent trajectories generated by the same skill z. When different offsets are applied, the same skill fails to reproduce consistent trajectories, providing clear evidence of brittle generalization.
> > >
> > > **3. Revised Paragraph for Section 4.1.1 (Clarifying the role of EMA)**
> > > To eliminate any misunderstanding regarding the EMA, we have reordered the paragraphs and separated the implementation details using a sub-heading (**lines 172-182** of the submitted paper):
> > >
> > > > Iterating the ($z$-step $\rightarrow$ $\phi$-step) mitigates the gradient conflicts caused by a fixed $z_{roll}$, thereby resolving the aforementioned semantic drift. We demonstrate that our iterative update is mathematically robust and structurally stable at each step (Details in Appendix A).
> > >
> > > > **Implementation details of skill relabeling:** In practice, to reduce training instability, we compute the relabeled targets using an exponential moving average (EMA) network [Schwarzer et al., 2020], as shown in Equation (4).
> > >
> > > **4. Revised Algorithm 1**
> > > We have comprehensively updated Algorithm 1 to resolve typos and readability issues:
> > > 1) Corrected notation inconsistencies with the main text (e.g., $\pi(a \mid s, z) \rightarrow \pi(a \mid \ell, z)$).
> > > 2) Improved the mathematical readability of the skill relabeling process (e.g., explicitly formalizing it as $x = (s, s', z_{roll}) \sim \mathcal{B}$ and relabeling to $x_{relab} = (s, s', z_{relab})$) to readability.
> > > 3) Fixed line-break issues that previously hindered readability.
> > > 4) Corrected the typo where the output of the CIB $q_{\psi}(\ell \mid s)$ was incorrectly denoted as $l$ instead of $\ell$.
> > >
> > > **5. Clarification of "Data-efficient"**
> > > We identified that “sample efficiency” and “data efficiency” were used interchangeably in the Introduction and Experiment sections. To ensure consistency with the title, we unify these terms to “data efficiency.”
> > >
> > > Additionally, to better highlight the connection between our experimental results and data efficiency, we created a separate subsection in **Section 5.3** focusing on the findings from **lines 319-323** of the submitted paper:
> > >
> > > > **Data-efficiency.** To evaluate the data efficiency of GenDa, we measure the interactions required to match the asymptotic performance of baselines trained for 10M steps in the Humanoid-Numeric environment. Remarkably, GenDa achieves a state coverage of 1000 with 5x fewer interactions (2M steps). Moreover, Table 2 shows that the downstream task performance of GenDa trained for only 2M steps surpasses that of baseline algorithms trained for 10M steps. This superior sample efficiency positions GenDa as a highly practical foundation for real-world robotic learning, where data collection is inherently costly."
> > >
> > >
> > > We hope these concrete revisions fully assure you that your concerns have been meticulously addressed. Please let us know if there are any further adjustments you would like to see.

---

### Official Review · Reviewer_QUqV · 2026-03-10

**Soundness:** 3
**Presentation:** 3
**Significance:** 3
**Originality:** 3
**Overall Recommendation:** 5
**Confidence:** 4

**Summary:**

This paper studies unsupervised reinforcement learning for skill discovery, focusing on improving data efficiency and generalization of skill policies. The authors identify two key issues in existing off-policy URL methods: semantic drift of skill labels in replay buffers and poor generalization due to dependence on global contextual information. To address these problems, they propose GENDA, which combines a skill relabeling mechanism that recomputes skill labels based on the current latent representation and a Complementary Information Bottleneck (CIB) that encourages policies to rely on ego-centric features rather than global context. Experiments across several control environments show improved state coverage, downstream task performance, and sample efficiency compared to prior skill discovery methods.

**Compliance With Llm Reviewing Policy:**

Affirmed.

**Final Justification:**

The rebuttal addressed all my concerns, and therefore I'm raising my scores.

**Key Questions For Authors:**

1. How sensitive is GENDA to the quality of the learned representation \phi?
2. Does the CIB module consistently improve performance across environments, or only in tasks with strong global contextual signals?
3. How do you mix the different z? How important is it to mix them rather to use the fully relabeled z?
4. The baseline methods in Table 1 has near-0 performance in many environments. Is this expected? Why is this?

**Limitations:**

yes

**Strengths And Weaknesses:**

strengths:
- Presentation is clear the the problems are well motivated. The paper clearly identifies semantic drift in replay buffers and generalization as key issues in off-policy unsupervised RL.
- The proposed method is clear and easy to intuitively understand
- The empirical results are strong, showing that the proposed method significantly outperform baselines.

Weaknesses:
- The effectiveness of relabeling relies heavily on the quality of the latent representation, which could affect the performance of the method
- The empirical results heavily focus on locomotion domains, and lack of manipulation tasks
- While the two identified limitations are intuitive to understand, they are not shown empirically

---

> ### Author Rebuttal · Authors · 2026-03-25
>
> We sincerely thank the reviewer for the constructive feedback and for recognizing the clarity, strong empirical results, and intuitive appeal of our proposed method. We address your questions and concerns below.
>
> **Relabeling dependency on quality of the $\phi$ latent representation:**
> To clarify, our skill relabeling does not passively rely on the quality of $\phi$. Instead, the representation quality improves in a self-predictive manner through our skill relabeling.
> As detailed in **Section 4.1.1**, we employ an skill relabeling mechanism, where past trajectories are reinterpreted using the current latent function and telescoping sum. By updating the representation learning labels with current semantics, GenDa mitigates semantic drift that hinders stable learning.
> This creates a **virtuous cycle**: stabilized learning improves $\phi$'s latent representation. Thus, GenDa is not highly sensitive to $\phi$'s representation quality, as the skill relabeling inherently drives it toward high quality.
>
> **Empirical evidence for the limitations:**
> * **Semantic Drift:** In off-policy learning, freezing $z$ in the replay buffer means the same $z$ induces drastically different trajectories depending on the policy's training stage, as explicitly shown in **Figure 1**. This variance directly contradicts and destabilizes the representation learning objective.
> * **Overfitting to Global Context:** **Figure 2** demonstrates how baseline skill policies overfit to specific global state information. Offset(a,b) denotes that xy-position of init-state is shifted to (a,b) coordinates. The same colored curves represent physical trajectories generated by the same skill. The agent fails to reproduce the same coverage for the same skill when given offset is varied, proving brittle generalization.
>
>
> **Lack of manipulation tasks:**
>  The focus on locomotion stems from the fundamental nature of metric-aware URL objectives, which are designed to maximize diverse, temporal state changes. Locomotion provides an unbounded space to clearly measure this state coverage scalability, whereas manipulation tasks involve fixed-base agents with naturally bounded state coverage ceilings. We will clarify this distinction in the revised text.
>
> **Does CIB consistently improve performance across environments?:**
>  Based on our analysis, we believe the CIB module can consistently serve as an effective regularizer for generalization across a wide variety of environments.
> We attribute the root cause of policy entanglement in **Section 3.2** to a "optimization shortcut" behavior in reinforcement learning. In the standard URL framework, the skill policy is optimized to maximize an intrinsic reward derived from the representation function $\phi$. If the specific information modeled by the skill $z$ is redundantly present in the raw state observation $s$, the policy $\pi(a \mid s, z)$ naturally exploits this redundant context as a shortcut to easily maximize the RL objective.
>
> The CIB module directly prevents the skill policy from falling into this reward-maximizing shortcut. By structurally filtering out the redundant context from the observation of policy $\pi(a \mid \psi(l \mid s),z)$, it forces the agent to rely purely on the semantic meaning of $z$. We conclude that CIB can consistently function as a vital regularization mechanism for robust generalization across most environments.
>
> **Mixing different $z$ and its importance:**
> In Appendix C, we provide detailed explanations on how we mixed different $z$ values, along with the empirical results of using various mixtures to generate $z$ for policy relabeling. As illustrated in **Figure 10 in the Appendix**, the strategy of mixing $z$ yields superior performance.
>
> Specifically, when comparing our method (which utilizes mixing) to fully relabeled cases, our approach outperformed both $c$ (green) and Mix($c$, $\tau$) (blue). While the fully relabeled $\tau$ (red) achieved comparable mean performance, our mixing strategy demonstrated a significantly lower standard deviation in state coverage across seeds (Ours: 123 vs. $\tau$: 370). Due to this superior stability and performance, we adopted and fixed this mixing strategy as our primary approach.
>
> **Near-zero baseline performance in Table 1:**
> The near-zero performance of baselines in **Table 1** is expected and stems directly from the two bottlenecks we identified:
> 1.  **Failure to learn skills (**Section 3.1** ):** In more challenging environments such as Dog and Fish, baselines fail to discover meaningful skills during the pretraining phase (**Figure 3e, 3f**).
> 2.  **Failure to generalize (**Section 3.2** ):** Although baselines learn meaningful skills and achieve reasonable performance in -FSRG, failure in -RSRG and -Maze stems from their policies are overfitted with the global context.

---

> > ### Author Rebuttal · Reviewer_QUqV · 2026-04-01
> >
> > Thanks to the authors for their response. This fully addresses my concerns and I have raised my scores accordingly.

---

> > > ### Author Response · Authors · 2026-04-04
> > >
> > > Thank you for your careful reading and constructive feedback. We are pleased that our response addressed your concerns. We will incorporate the relevant clarifications into the revised version of the paper. Thank you again for raising your score.

---

### Official Review · Reviewer_6Gkd · 2026-03-10

**Soundness:** 3
**Presentation:** 3
**Significance:** 2
**Originality:** 3
**Overall Recommendation:** 4
**Confidence:** 4

**Summary:**

This work points out that off-policy unsupervised reinforcement learning faces two major challenges: sample inefficiency due to non-stationary skill semantics and brittle generalization caused by overfitting to global contexts. To address these challenges, this work proposes GenDa (Generalizable Data-efficient Agent) with two main components: Skill Relabeling Mechanism and Complementary Information Bottleneck. Empirical results show that GenDa achieves superior state coverage and succeeds in challenging, high-dimensional environments (e.g., Dog-Numeric) where existing methods fail.

**Compliance With Llm Reviewing Policy:**

Affirmed.

**Final Justification:**

Thanks to the authors for their response. All my concerns have been addressed, and I vote for acceptence to this work.

**Key Questions For Authors:**

-  Q1: Are there any ablation studies of $\beta$?

- Q2: What if an episode contains heterogeneous behaviors or rapid reactive adjustments?

Overall, I keep positive about this work and think it will raise the interest of the community. I will keep active in the following discussion stage and change my score accordingly.

**Limitations:**

This work has discussed Open Challenges and Future Directions in Sec. 7.2.

**Strengths And Weaknesses:**

Strengths:

- The paper directly solves two overlooked bottlenecks in off-policy URL: non-stationary skill semantics (semantic drift) and brittle generalization caused by overfitting to the global context.

- This paper is well written and easy to follow.

- Extensive experiments in challenging benchmarks demonstrate the superior sample efficiency of GenDa.

Weaknesses:

- The proposed method introduces some extra hyperparameters like $\beta$ to balance two losses, which will make it more complicated.

- The skill relabeling mechanism seems to rely on the assumption that a single skill is executed per episode.

- Minors: in the caption of Fig.2, (Park et al 2024) is not a noun, it is better to use \citeauthor.

- These are two major classes of methods for unsupervised RL: unsupervised exploration and skill discovery. Although this work focuses on skill discovery, a thorough discussion and comparison with exploration-based methods [1-4] is necessary.

Reference:

[1] Unsupervised reinforcement learning in multiple environments

[2] Rethinking exploration in reinforcement learning with effective metric-based exploration bonus

[3] Exploratory Diffusion Model for Unsupervised Reinforcement Learning

[4] Latent exploration for reinforcement learning

---

> ### Author Rebuttal · Authors · 2026-03-25
>
> We sincerely thank the reviewer for the positive assessment of our work, highlighting its readability, extensive experiments, and effectively addressing the core bottlenecks in off-policy URL. We address your questions and feedback below.
>
> **Ablation study of the uniformity coefficient $\beta$:**
>  We completely agree that GenDa's scalability would be hindered if the newly introduced hyperparameter $\beta$ required heavy, environment-specific tuning. To dispel this concern, we conducted additional skill pre-trianing experiments across diverse environments with 3 seeds using various values of $\beta$ to verify its robustness.
> We will add this ablation results on the experiments section of revised text.
>
> | $\beta$ | Humanoid-Numeric(10M) | Quadruped-Numeric(5M) | Quadruped-Pixel(3M) | Humanoid-Pixel(5M) |
> | :--- | :--- | :--- | :--- | :--- |
> | **0.1** | 2094±89 | 1740±223 | 286±85 | 127±72 |
> | **0.5** | 2236±103 | 1597±66 | - | - |
> | **1.0 (ours)** | 2192±112 | 1794±220 | 331±75 | 124±37 |
> | **2.0** | 2213±192 | 1531±409 | - | - |
> | **4.0** | 2595±386 | 1783±63 | 285±77 | 123±30 |
>
> *(Because of the computational limitation, we use two options(0.1 and 4.0) for the $\beta$ ablation in "-Pixel" environments.)*
>
> The results highlight that while performance varies depending on $\beta$, the model is not hyper-sensitive to the point where slight variations trigger an objective collapse in either numeric or pixel domains. This provides evidence that our proposed objective, driven by skill relabeling and the uniformity regularizer, can be deployed across varied environments without requiring exhaustive per-environment tuning.
>
> **Handling heterogeneous behaviors or rapid reactive adjustments:**
> This is a very insightful point. In this work, we followed the standard USRL pretraining protocol where a single skill $z$ is sampled and fixed for the entire episode. This naturally biases the collected trajectories to exhibit homogeneous behaviors.
>
> However, as we noted in our future work section, we completely agree that handling heterogeneous behaviors and rapid reactive adjustments is a crucial next step, especially for complex downstream tasks or learning from offline, multi-intent datasets.
>
> While we did not explicitly propose a method for *segmenting* episodes in this paper, our skill relabeling framework is structurally and mathematically equipped to handle intra-episode skill changes. Because our relabeling target is derived from the telescoping sum $\sum (\phi(s_{t+1}) - \phi(s_t))$, it can be generalized to any arbitrary time segment $[t_1, t_2, ..., t_n]$. Therefore, if a heterogeneous episode is segmented into sub-trajectories based on behavioral intent, our method can safely and mathematically consistently relabel those segments without breaking the learning objective. We will expand on this theoretical compatibility in the discussion section.
>
> **Response to Weaknesses and Minor points:**
> * **Comparison with Exploration-based methods:** We agree that a thorough discussion of exploration-based methods is necessary to provide a complete picture of the URL landscape. While exploration methods (like those in the suggested references) excel at directly maximizing state entropy and novel visitations, skill discovery methods (like GenDa) focus on abstracting these visitations into controllable, reusable behavioral primitives (skills) for hierarchical downstream tasks. We will add a dedicated paragraph in the Related Work section discussing this distinction and citing the valuable papers you provided.
> * **Citation formatting:** Thank you for catching the typo in the caption of Figure 2. We will correct the citation format to use `\citeauthor` properly in the revised manuscript.

---

> > ### Author Rebuttal · Reviewer_6Gkd · 2026-04-03
> >
> > Thanks to the authors for their response. All my concerns have been addressed, and I vote for acceptence to this work.

---

> > > ### Author Response · Authors · 2026-04-04
> > >
> > > Thank you for your careful reading of our paper and for your thoughtful and constructive feedback. We are pleased that our responses helped address your concerns. We will reflect these points in the revised version of the paper. Thank you again for your positive evaluation.

---

### Official Review · Reviewer_CsFq · 2026-03-13

**Soundness:** 3
**Presentation:** 3
**Significance:** 3
**Originality:** 3
**Overall Recommendation:** 4
**Confidence:** 3

**Summary:**

This paper addresses two critical bottlenecks in off-policy unsupervised reinforcement learning (URL): sample inefficiency caused by semantic drift and brittle generalization resulting from global context overfitting. The authors propose the GenDa framework, which utilizes a dynamic skill relabeling mechanism to correct stale labels in the replay buffer, and a Complementary Information Bottleneck (CIB) to disentangle ego-centric state information from global contextual signals.

**Compliance With Llm Reviewing Policy:**

Affirmed.

**Final Justification:**

The authors’ rebuttal has effectively addressed my primary concerns and provided meaningful clarifications on several key aspects of the work. In particular, the additional discussion on the limitations in manipulation tasks (e.g., Kitchen) and the newly provided quantitative evaluation under visual distribution shifts strengthen the empirical support for the proposed method.

Though some limitations remain—such as performance variability in complex interaction environments and the still limited robustness under broader distribution shifts—these issues are now clearly acknowledged and better contextualized. Importantly, these limitations do not undermine the core contributions of this work.

Overall, I consider this a technically solid and well-motivated work. Also, the rebuttal has addressed my concerns and reinforced my assessment. I therefore maintain my recommendation of Weak Accept (4).

**Key Questions For Authors:**

1. Since β controls the trade-off between skill alignment and representation diversity, could the authors provide an ablation study showing how different values of β affect performance across environments?
2. If the metric-learning objective leads to capturing local proprioceptive information useful for control, does minimizing mutual information risk remove useful signals for policy learning?
3. In Section 4.2, the method mixes several z values. How sensitive is the performance to the mixing strategy or ratios?
4. The Dog-Numeric environment is a clear success case for GenDa. Why do the authors think prior methods fail in this environment while GenDa succeeds?

**Limitations:**

Yes

**Strengths And Weaknesses:**

Strengths: 1) Identifying the semantic drift problem caused by stale skill labels in replay buffers in off-policy URL.  2) Proposing a skill relabeling technique paired with a contrastive uniformity regularizer to stabilize representation learning and maintain skill diversity without representation collapse. 3) Introducing the CIB module to decouple policy inputs from global contextual information, improving robustness and generalization. 4) Demonstrating strong empirical performance across multiple benchmarks, notably succeeding in the highly complex 226-dimensional Dog-Numeric task, where prior methods struggle.

Weaknesses: 1) Despite strong performance in locomotion tasks, GenDa exhibits high variance and only marginal performance gains over baselines (e.g., CIC) in the Kitchen-Pixels task (Figure 3(g)). This instability in complex object-interaction environments weakens the claim that it provides a general-purpose foundation for control. 2) The paper lacks an explicit quantitative evaluation under visual distribution shifts to directly demonstrate the effectiveness of the CIB module. While Figure 8 provides a qualitative visualization of representation disentanglement, experiments with controlled visual perturbations (e.g., background or texture changes) would better support the robustness claim.

---

> ### Author Rebuttal · Authors · 2026-03-25
>
> We sincerely thank the reviewer for the positive assessment of our work, for recognizing our formalization of the critical bottlenecks in off-policy URL and the strong empirical performance of our proposal. We address your questions and feedback below.
>
> **Marginal performance in manipulation**: We agree that Kitchen is a weak case for GenDa. We believe this is because metric-aware URL objectives are naturally aligned with locomotion tasks that allow broad temporally extended coverage, whereas Kitchen is a fixed-base manipulation domain with a much shorter episode length of 50 steps.
>
> We believe the instability in Kitchen is partly due to the task structure: success depends on whether exploration discovers useful interaction sequences, which can amplify seed-to-seed variance. We will clarify this limitation and tone down our “general-purpose foundation” claim accordingly.
>
> **Qualitative results of CIB in visual environment**:
>  We fully agree that a quantitative evaluation under visual distribution shifts is crucial to comprehensively demonstrate the effectiveness and operational range of the CIB module's robustness.
>
> To address this, we conducted an additional evaluation in the Quadruped-Pixels environment across 4 seeds. We tested the skill policy's robustness by applying unseen visual shifts to the background(Original: **Figure 11 (f)** in the Appendix).
>
> 1. **Unicolor:** 4 distinct solid color backgrounds.
> 2. **Gradation:** 4 distinct gradient patterns.
>
> The quantitative results for state coverage are in the table below:
> | | Ours | Ours(w/o CIB) |
> | :--- | :--- | :--- |
> |**Original**| 331±75 | 335±103 |
> | **Unicolor** | 268±25(-19%) | 137±99(-59%) |
> | **Gradation** | 206±88(-38%) | 78±47(-77%) |
>
> Consequently, the results demonstrate that the CIB provides relative robustness compared to the w/o CIB in pixel environments, but its effectiveness is degraded under the visual shifts.
>
> We abstained from using any image augmentation techniques during visual training to ensure a fair comparison with the baselines. This highlights a highly promising direction: because the CIB module structurally disentangles redundant context, incorporating standard visual generalization techniques like [1] during CIB training would seamlessly bridge this gap and yield a model highly robust to visual characteristics.
> We will include this experiment, and the accompanying discussion.
>
> **Ablation study of the uniformity coefficient $\beta$**:
>  We apologize that strict character limits prevent us from detailing the answer here. Please refer to our response to Reviewer 6Gkd, who raised a similar point.
>
> **CIB and useful signal loss**:
> We believe the CIB is safe from such risks because our objective $\mathcal{J}_{CIB}$ is structured to act as a complementary filter rather than a destructive one. While the penalty term $-I(l;\phi(s))$ minimizes the redundant information already captured by the $\phi(s)$, the maximization term($I(l;s)$) encourages the CIB to preserve remaining information in $s$.
> Therefore, the policy$\pi(l, z)$ still has access to the necessary control signals while safely avoiding overfitting to the redundant context.
>
> **Success in Dog-Numeric**: The Dog-Numeric environment is highly challenging primarily due to its massive state dimension (226 dims), which inherently introduces a very high degree of state variance. We believe prior off-policy methods fail here because they suffer from a compound variance effect. As discussed in **Section 3.1** of our paper, using static $z$ labels in an off-policy setting injects high-variance noise into the learning process. In a high-dimensional environment like Dog, this artificial noise from fixed labels exacerbates the natural variance of the state space. GenDa succeeds because our skill relabeling mechanism directly mitigates this semantic drift. By dynamically correcting the labels, we reduce the variance associated with off-policy learning. This allows GenDa to focus entirely on overcoming the inherent complexity of the Dog environment itself.
>
> **Mixing strategy for skill policy relabeling**:
> In **Appendix C**, we provide detailed explanations on how we mixed different $z$, along with the empirical results of mixing strategy (**Figure 10** in Appendix). We employed Mix($z_c, z_{relab}, z_{roll}$) because of its performance and stability across seeds.
>
>
> We also tested various mixing hyperparameters for GenDa in Quadruped-Numeric(3 seeds), and observed marginal performance differences. Results are summarized in the table below:
>
> | $c-\tau$ mixture ratio | State Coverage |
> | :--- | :--- |
> |**0.5:0.5(original)**| 1851±123 |
> |**0.25:0.75** | 1820±241 |
> |**0.75:0.25** | 1842±229 |
>
> | Threshold for $z_{roll}$ | State Coverage |
> | :--- | :--- |
> |**0.4(original)**| 1851±123 |
> |**0.8** | 1817±245 |
> |**0.95** | 1678±318 |
>
> **Reference**
>
> [1] Yarats, Denis, et al. "Mastering visual continuous control: Improved data-augmented reinforcement learning." arXiv preprint arXiv:2107.09645 (2021).

---

> > ### Author Rebuttal · Reviewer_CsFq · 2026-04-04
> >
> > Thanks for the detailed rebuttal. The additional explanations help me better understand this work. I am maintaining my positive rating of this paper.

---

> > > ### Author Response · Authors · 2026-04-08
> > >
> > > We sincerely appreciate your thorough review of our manuscript and your valuable, constructive feedback. We are glad to hear that our clarifications successfully resolved your initial concerns. Thank you once again for your positive assessment.

---

### Decision · Program_Chairs · 2026-04-30

**Decision:**

Accept (regular)

**Comment:**

This paper studies unsupervised reinforcement learning (URL) for skill discovery, focusing on improving data efficiency and generalization of skill policies. The authors identify two key issues in existing off-policy URL methods: semantic drift of skill labels in replay buffers and poor generalization due to dependence on global contextual information. To address these problems, they propose GenDa (Generalizable Data-efficient Agent), which combines a skill relabeling mechanism that recomputes skill labels based on the current latent representation and a Complementary Information Bottleneck (CIB) that encourages policies to rely on ego-centric features rather than global context. Experiments across several control environments show improved state coverage, downstream task performance, and sample efficiency compared to prior skill discovery methods.

Reviewers note that the paper is well-written and easy to follow (6Gkd). Presentation is clear and the problems are well motivated. The proposed method is clear and easy to intuitively understand (QUqV, a9e4, 6Gkd). Extensive experiments in challenging benchmarks demonstrate the superior sample efficiency of GenDa, showing that the proposed method significantly outperform baselines (6Gkd, QUqV).

The reviewers raise several weaknesses in their reviews, which the authors successfully address in the rebuttal responses. The authors have acknowledged certain limitations, clarified certain misconceptions in understanding, and conducted additional experiments to show the robustness of their approach. They have promised to update the camera ready version of the paper with the discussion points and new results that emerged in conversation with the reviewers (the detailed list is available in response to reviewer a9e4).
- The authors acknowledge the weakness raised by CsFq and QUqV about the empirical results being heavily focused on locomotion tasks. The authors note that metric-aware URL objectives are naturally aligned with locomotion tasks. They promised to clarify this limitation and tone down their “general-purpose foundation” claim accordingly.
- CsFq points out that the paper lacks an explicit quantitative evaluation under visual distribution shifts to directly demonstrate the effectiveness of the CIB module. The authors agree and to demonstrate the CIB module's robustness conducted an additional evaluation in the Quadruped-Pixels environment across 4 seeds, which was convincing to the reviewer.
- 6Gkd mention that the proposed method introduces some extra hyperparameters, and the need to tune them can make the approach more complicated. The authors conducted an experiment to provide evidence that their proposed objective can be deployed across varied environments without requiring exhaustive per-environment tuning.
- The authors will add a dedicated paragraph in the Related Work section discussing the distinction between their work with exploration-based methods (another class of unsupervised RL methods), citing some missing references provided by 6Gkd.
- a9e4 suggested that some parts of the paper writing could be improved, such as the presentation of Algorithm 1 and Sec 4.1.1.  The authors will formalize equations more clearly, thoroughly revise typos and formatting, and clarify the misunderstanding in Sec 4.1.1 that the EMA itself resolves semantic drift.
- 6Gkd notes that the skill relabeling mechanism seems to rely on the assumption that a single skill is executed per episode. The authors note this is a reasonable direction for future work, and that their approach can be generalized to any arbitrary time segment, if segments are available. They will expand on this theoretical compatibility in the discussion section.

All reviewers' concerns were fully resolved by the rebuttal responses. Overall, all reviewers maintain or increase their rating of the paper towards acceptance. I recommend the paper for acceptance given the authors will include all the changes approved by reviewers in the camera-ready version of the paper.